# Expert Views on the Future Development of Biogas Business Branch in Germany, The Netherlands, and Finland until 2030

**Erika Winquist** [1,*]**, Michiel Van Galen** [2]**, Simon Zielonka** [3]**, Pasi Rikkonen** [1]**, Diti Oudendag** [2]**, Lijun Zhou** [3] **and Auke Greijdanus** [2]

1   Natural Resources Institute Finland (Luke), P.O. Box 2, 00791 Helsinki, Finland; pasi.rikkonen@luke.fi
2   Wageningen Economic Research, Wageningen University & Research, Prinses Beatrixlaan 582-528, Postbus 29703, 2502 LS Den Haag, The Netherlands; michiel.vangalen@wur.nl (M.V.G.); diti.oudendag@wur.nl (D.O.); auke.greijdanus@wur.nl (A.G.)
3   State Institute of Agricultural Engineering and Bioenergy, University of Hohenheim, 70599 Stuttgart, Germany; simon.zielonka@uni-hohenheim.de (S.Z.); Lijun.Zhou@uni-hohenheim.de (L.Z.)
*   Correspondence: erika.winquist@luke.fi

**Abstract:** To be able to meet the European Union's energy and climate targets for 2030, all member states need to rethink their energy production and use. One potential renewable energy source is biogas. Its role has been relatively small compared to other energy sources, but it could have a more central role to solve some specific challenges, e.g., to reduce carbon dioxide ($CO_2$) emissions from traffic, or to act as a buffer to balance electricity production with consumption. This research analyses how the future of the biogas business in three case study countries is developing until 2030. The study is based on experts' views within the biogas business branch in Germany, The Netherlands, and Finland. Both similarities and differences were found among the experts' answers, which reflected also the current policies in different countries. The role of biogas was seen much wider than just to provide renewable energy, but also to decrease emissions from agriculture and close loops in a circular economy. However, the future of the biogas branch is much dependent on political decisions. To be able to show the full potential of biogas technology for society, stable and predictable energy policy and cross-sector co-operation are needed.

**Keywords:** expert survey; renewable energy; biogas; biomethane; biogas plant; business model; political support system

## 1. Introduction

Renewable energy production is growing fast in the European Union (EU) and globally. In the first half of 2020, renewable electricity generation in the EU exceeded fossil fuel generation for the first time ever. This was partly due to the 7% fall in electricity demand because of the coronavirus (COVID-19) pandemic. However, especially the electricity generation using wind and solar energy has grown over a longer time period, from 13% of total electricity generation in 2016 to 21% in the first half of 2020 [1].

The development is most welcomed because the EU aims to be carbon neutral by 2050 [2]. Recently, even China announced to have $CO_2$ emissions peak before 2030 and achieve carbon neutrality before 2060 [3]. China's commitment is crucial when mitigating climate change as it is responsible for around 28% of global emissions. Finnish emissions might be less important globally, but the goal is even more ambitious; Finland aims to achieve carbon neutrality by 2035 [4].

Individual EU member countries have varying targets for the share of sustainable energy sources and various ways of achieving their renewable energy targets. Replacing fossil fuels with renewable energy sources is, however, much more than just switching the fossil raw materials to renewable ones. Unlike many renewables, fossil fuels are flexible to use in versatile applications and easy to store. Thus, the whole energy system needs

to be built again on a renewable basis [5]. Instead of few large energy sources, several energy sources are integrated in the renewable system [6]. Moreover, instead of centralised solutions, the energy is produced locally [7].

The new renewable energy system must tackle several problems—how to balance electricity production with consumption, how to arrange the energy needed for traffic, and how to ensure local energy security and affordability. Biogas could provide solutions to each of these questions, although not alone because biomass resources are limited. In addition to biogas and biomethane obtained by upgrading biogas, corresponding renewable alternatives to natural gas can be produced by power-to-gas from hydrogen produced with renewable electricity and $CO_2$ captured from industrial processes, and synthetic natural gas (SNG) from biomass gasification [8]. These can both increase the production potential of biomethane and use the same existing infrastructure as natural gas.

The German energy transition (Energiewende) was the first attempt to transform a centralised fossil-based energy system into a local renewable-based system. The generous feed-in tariffs (FiT) enabled renewable electricity production, especially from biogas. Today, Germany is the world leader, with 9527 biogas plants by the end of 2019 [9] and a 13.0% share of biogas/biomethane in renewables-based electricity generation [10]. However, by changing the FiTs to a tendering based system, the current subsidy system favors wind and solar over biogas and large production facilities over small ones [11]. To be able to maintain the production, the biogas plants need to find cost savings, improvements in energy efficiency, and new business models.

This research focusses on the biogas business and its prospects toward 2030. Based on experts' views in the biogas and energy branch in Germany, The Netherlands, and Finland, this research analyses how the future of the biogas business in three case study countries is developing until 2030. By using an expert survey method, expert views of the future are used to map the probable and desirable future views.

The research questions are the following:

- How is the business environment of the renewable energy production evolving until 2030 in the case study countries and the EU?
- How do experts see the probable and desirable future paths of the use of biogas and its role in the energy transition towards renewable energy?
- Which income sources will be more significant in the future for the biogas business branch?

## 2. Background

Biogas is a mixture of methane (50–70%) and carbon dioxide (30–50%), whereas natural gas is almost pure methane ($CH_4$). However, also biogas can be upgraded to biomethane (>92% $CH_4$). Biogas is formed when micro-organisms degrade organic compounds in anaerobic conditions and the process is called anaerobic digestion (AD). Biogas is still collected from old landfill areas, but thanks to the Landfill Directive (1999/31/EC), which obliges the member states to reduce the amount of biodegradable municipal waste that they landfill to 35% of 1995 levels by 2016, the volumes are decreasing. On the contrary, biogas production in reactor plants is increasing. Suitable raw materials are municipal sewage sludge and biowaste, side streams from the food and paper processing industries, as well as from agriculture such as manure, grass, straw, and other crop residues. Biogas can be used for the production of heat, combined heat and power (CHP), and upgraded to biomethane, which can be used as traffic fuel or to replace natural gas in various applications.

### 2.1. Overview of Current Biogas Production in Europe

The biogas production in the European Union represents roughly half of the global biogas production [12]. The relative importance of biogas in the EU is mainly thanks to Germany which represents half of the EU production. Germany was one of the first European countries to implement a subsidy for renewable electricity and biogas production. Already in 1991, the Electricity Feed-In Law was introduced, and in 2000, the Renewable

Energy Sources Act (EEG, or Erneuerbare Energien Gesetz). The EEG had several updates (2004, 2009, 2012, and 2014) before, in 2017, the basis for its support changed fundamentally to an auction model with lower maximum achievable tariffs. In addition, due to the strong position of Germany in the European biogas production, the political changes in Germany reflects the whole EU level.

The change in subsidy levels can be seen also in the biogas production development in Germany (Figure 1). Until 2015, there was strong growth, but after that, the production has stayed at the same level. The biomethane production data was not easily available. Thus, the biomethane addition to the natural gas network (or use as traffic fuel in the case of Finland) has been used as an indicator of the development of the biomethane market (Figure 1). Biomethane addition to the gas grid decreased somewhat in Germany in 2017. However, according to the EBA report published in 2020 [13], five new biomethane upgrading plants were built in 2018, which would indicate that the interest in biomethane upgrading is still increasing.

In The Netherlands, biogas production has continued the slow growth, whereas biomethane addition to the natural gas network is growing fast (Figure 1). In Finland, there was a stepwise growth in biogas production in 2017, but after that year, the production has stayed at the same level. In 2016, the state-owned Gasum Ltd. entered the market, buying two companies with seven biogas plants and becoming the largest biogas producer in Finland [14]. The same year, Gasum also built one additional larger biogas plant and started biogas upgrading to biomethane, as well as expanding the gas filling station network. The use of biomethane as traffic fuel has continued growing after that (Figure 1).

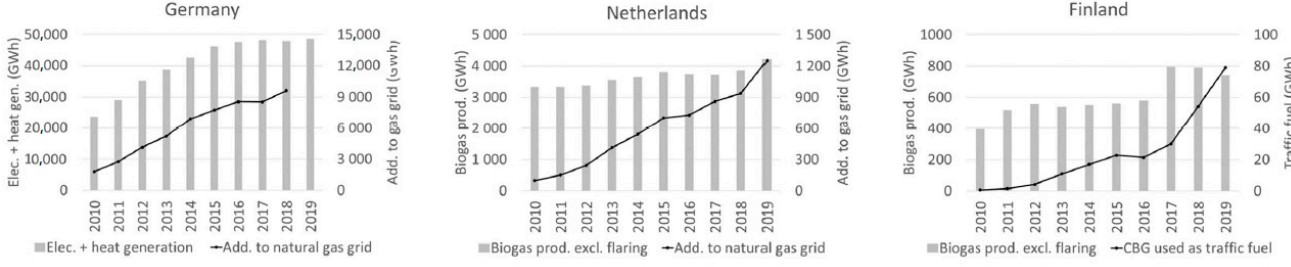

**Figure 1.** Primary axis: production of electricity and heat from biogas (Germany) [15,16]/biogas production excl. flaring (The Netherlands, Finland) [17,18]; secondary axis: addition of biomethane to the natural gas grid (Germany, The Netherlands) [19–21]/use of biomethane as traffic fuel (Finland) [22].

Biogas consumption and production potential in the case study countries can be estimated both with the natural gas consumption, which describes the existing infrastructure also available for biomethane use, and the availability of agricultural biomasses, which provide the largest raw material reserve for biogas production. In all case study countries, the estimated biogas production represents only a fraction of the natural gas consumption (Table 1). Especially in The Netherlands, natural gas consumption is at a high level compared to the population. However, because of safety reasons for the inhabitants of the province of Groningen, where the production of natural gas has caused earthquakes, the aim is to phase out natural gas until 2050 [23]. The focus in gas transition is on energy savings, replacing natural gas with biomethane or hydrogen gas, producing heat with other renewable energy options, and using natural gas only as feedstock for the chemical industry [24].

Although Finland has nearly the same total area as Germany, the cultivated area in Finland is at the same level as in The Netherlands (Table 1). The cultivated area correlates with the availability of energy crops for biogas production but also with side streams from other crop production such as grass cultivated as green manure, catch and cover crops, straw, and crop residues. Despite the rather limited cultivated area in The Netherlands, the number of livestock is high, and thus the estimated biomethane potential in the collectable manure is almost half of that in Germany (Table 1). If only manure biomethane potential

is considered, both The Netherlands and Finland could double their current biogas production. Only in Germany, manure biomethane potential is (22,130 GWh) just half of the current production (48,747 GWh) (Table 1) [25].

Energy crops covered nearly half of the biogas raw material supply in Germany in 2018. Almost the same share is covered by agricultural residues (including manure), and only some biowaste/municipal waste (i.e., organic fraction of municipal solid waste) and industrial side streams (food and drink) are used [13]. Previously, the share of energy crops, particularly maize, has been even larger in Germany, but the utilisation of maize silage and corn has now been limited since the EEG 2017. Initially, from the beginning of 2017, maize was limited to a maximum of 50% for the mass-based substrate input, then later to 47% in 2019–2020 and further to 44% in 2021–2022 [26].

In The Netherlands, when the landfill plants are excluded, the produced biogas originates from biowaste/municipal waste (ca. 40%), co-digestion of agricultural/municipal/industrial side streams (ca. 40%), and sewage sludge (ca. 20%) [27]. In Finland, biogas production relies strongly on biowaste/municipal waste (ca. 90%), the rest being sewage sludge (ca. 5%) and agricultural residues (ca. 5%) [13].

**Table 1.** Summary table of case study countries (data from 2019 unless reported otherwise).

|  | Germany | The Netherlands | Finland | Ref. |
|---|---|---|---|---|
| Population ($10^6$) | 83.0 | 17.3 | 5.5 | |
| Biogas production excl. flaring (GWh) | 48,747 [a] | 4210 | 740 | [10,17,18] |
| Biogas production per capita (kWh) | 587 [a] | 243 | 135 | [10,17,18] |
| Biomethane production (GWh) [b] | 10,292 | 1574 | 105 | [13] |
| Biomethane production per capita (kWh) [b] | 124 | 91 | 19 | [13] |
| Natural gas consumption (GWh) | 887,000 | 368,000 | 20,360 | [21,28] |
| Total area ($10^3$ km$^2$) | 357 | 42 | 338 | |
| Cultivated area ($10^3$ km$^2$) | 185 | 18 | 23 | |
| Collectable manure ($10^6$ t) [c] | 133 | 52 | 8 | [25] |
| Realistic manure biomethane potential (GWh) [c] | 22,130 | 9620 | 1250 | [25] |

[a] elec. + heat + vehicle fuel, excl. efficiency losses, [b] data from 2018, [c] data from 2013 (1 Nm$^3$ CH$_4$ = 10 kWh).

### 2.2. EU Level Directives and Goals

The aim of the original Renewable Energy Directive (2009/28/EC) was to promote renewable energy production in the EU, mitigate climate change, and increase the share of local energy production vs. imported fossil energy sources. The Renewable Energy Directive was revised in December 2018 (RED II) to better meet the emission reduction commitments under the Paris Agreement (December 2015) and to move the legal framework to 2030 [29]. Special emphasis in the revision is on the sustainability criteria for bioenergy, which will also include biomass and biogas for heating, cooling, and electricity generation. This will further steer the development from first-generation biofuels, where raw materials or land use is competing with food production, to second-generation biofuels exploiting various side streams and lignocellulosic raw materials.

Highlights of Revised Renewable Energy Directive (2018/2001/EU) [29] include the following:

- renewables should be 32% of the final energy consumption by 2030;
- national energy and climate plans (NECPs) for 2021–2030 (submitted to the European Commission by the end of 2019);
- renewable sources should account for 14% of transport fuels by 2030;
- strengthened sustainability criteria for bioenergy (including biomass and biogas for heating, cooling, and electricity generation);
- enabling self-production and -consumption of renewable energy;
- the original renewable energy directive will be replaced by 30 June 2021.

To meet the EU's energy and climate targets for 2030, EU member states need to establish a 10-year integrated national energy and climate plan (NECP) for the period

from 2021 to 2030. However, member states can decide the structure and scope of their plans individually. Some member states (e.g., Finland and France) brought up the role of biogas and biomethane, whereas others hardly mentioned it at all (e.g., Germany and The Netherlands) [13].

The next step in the EU's climate goals is carbon-neutrality by 2050, which is one of the targets in the European Green Deal initiative. The initiative includes all sectors of the economy and requires even higher greenhouse gas (GHG) emission reductions for 2030 than the RED II. Moreover, a European climate law was proposed in March 2020 to ensure to reach these goals [2]. In addition, to boost renewable energy and energy efficiency, the European Green Deal also contains the 'Circular Economy Action Plan' for sustainable industry and the 'Farm to Fork Strategy' for sustainable agriculture. Both initiatives open new possibilities for biogas, which provides renewable energy and the technology to use various side streams and cut emissions from agriculture.

Also linked to more sustainable agriculture, the European Regulation on Fertilizing Products (FPR) was approved in June 2019 [30]. The FPR recognises that fertilising products can be made from organic materials such as compost and digestate, and it establishes harmonised requirements to make them available on the internal market [13].

Furthermore, the 'Directive on the deployment of alternative fuels infrastructure' (2014/94/EU) was enforced in October 2014 [31]. The aim of this directive was to minimise the oil dependence of transport and reduce the environmental effects of transport throughout the EU. The national policy frameworks had to contain targets for alternative transport fuels and their distribution infrastructure, including pressurised gas fuelling points for 2020 and 2030. Although the requirements may be fulfilled with natural gas alone, biomethane can also be used.

*2.3. Governmental Support Systems and Goals*

2.3.1. Germany

In Germany, biogas production, as well as wind and solar electricity production, are supported through the latest version of the Renewable Energy Sources Act (EEG 2017), where the support system was switched from the feed-in tariffs (FiT) to the auction model (pay-as-bid). The newly built biogas plants with an installed electrical capacity of more than 150 kW$_{el}$ and already existing biogas plants can participate in auctions [26]. Power generation within the framework of the tender model of the EEG 2017 offers a financing possibility, especially for very cost-effective plants, which typically means very large plants. So far, the amount of electricity put out to tender is far from being used. This shows how tough the conditions of the tender are for plant operators. However, small units (up to a maximum of 100 kW), as well as liquid manure plants (higher FiT) and waste plants (FiT did not change), are still supported by a FiT support scheme [32]. The EEG Amendment 2021, which will come into effect on 1 January 2021, is sticking to the basic principle of making renewable power producers more market-oriented. Currently, the majority of the biogas plants in Germany, depend on the FiT, get lower price for feeding electricity to the grid than in their original EEG contract. The overall number of biogas plants is projected to face decreasing in 2020 for the first time [9].

Germany's Integrated National Energy and Climate Plan (NECP) contains the following goals to contribute to the achievement of the EU energy targets in 2030: (1) increasing energy efficiency by reducing primary energy consumption by 30% by 2030 compared to 2008, and (2) expansion of the share of renewable energies to 30% of gross final energy consumption in 2030. In addition, the NECP confirms the national GHG emission reduction target for 2030 of at least 55% compared to 1990, and the commitment to pursue GHG neutrality as a long-term goal by 2050 [33,34]. Specific biogas-related targets included in NECP are (1) 30% manure digestion by 2025 and (2) gas-tight storage of manure on up to 70% of biogas plants [35].

### 2.3.2. The Netherlands

The main support instrument for biogas and biomethane in The Netherlands is currently the so-called SDE++ regulation (Stimulering duurzame energieproductie en klimaattransitie), which has replaced a former regulation SDE+ that was active between 2013 and 2020. Both regulations, in addition to the earlier SDE regulation from 2008, intent to stimulate the production of renewable energy from all kinds of sources. The main difference in the new SDE++ regulation is that it is also applicable to projects that aim to reduce $CO_2$-emissions by, e.g., carbon capture or the use of excess heat from other sources.

The regulation in 2020 opens in four phases with an increased maximum amount of subsidy per reduced ton of $CO_2$. The subsidy compensates for the difference between the production costs of the renewable energy or $CO_2$ reduction technique and the market prices of the competing non-renewable energy (FiT of the non-profitable portion of production costs) for 12 to 15 years. The tariffs are guaranteed minimum income, which means that the scheme only pays out if energy prices are lower than the prices in the FiT for a certain category [26].

In The Netherlands, one type of tax allowance is currently relevant for renewable energy production from biomass—the ODE tax (Opslag Duurzame Energie- en Klimaattransitie), i.e., the surcharge for sustainable energy and climate transition paid on electricity and natural gas. Energy Investment Allowance (EIA scheme) is also available for eligible biogas installations in The Netherlands [13]. The SDE++ subsidy, however, cannot be used in combination with investment support due to EU restrictions on state support.

The biogas policies in The Netherlands are relatively stable in the sense that subsidies for biogas plants have continued in the so-called SDE regulation, now called SDE++. But there has been a significant shift in focus on agricultural biogas installations from co-fermentation of manure towards mono-fermentation of manure. The agricultural biogas production has grown considerably until 2011, but since then, the number of co-fermentation plants decreased. In 2019, there were 89 installations left [17]. The decrease in co-fermentation projects was due to high costs of coproducts and low electricity prices resulting from competition from solar and wind energy. In addition, government subsidies for co-fermentation have somewhat decreased, while projects that produce biomethane are stimulated. Mono-fermentation is less demanding in terms of management because no off-farm inputs have to be bought. Furthermore, new mono-fermentation installations have been developed and implemented that better suit the different scales of farms. In The Netherlands, a large dairy cooperative, FrieslandCampina, has been the driving force behind mono-fermentation at farm scale in the so-called Jumpstart-program. In 2020, there was considerable enthusiasm among dairy farmers to participate in the mono-fermentation project. One of the aspects of the approach is for farmers to lease the installation from the cooperative instead of having to entirely buy it themselves. In recent years, therefore, a shift can be observed from co-fermentation to mono-fermentation of manure on farms.

The target for 2030 in the Climate Agreement is to replace 70 PJ (19 TWh) of natural gas used by households and industry with 2 billion $m^3$ green gas. A roadmap for green gas was initiated in 2019. However, considering the current production, this will only be possible by developing several big gasification and digestion plants in the future [26]. In the roadmap, the Dutch government acknowledges that gas will play a role in the future energy supply. The roadmap predicts that 30–50% of the final demand for energy will consist of gas, either in the form of methane or hydrogen gas. A major driver for developing green gas is that the Dutch government has committed to decreasing the role of natural gas. In the Dutch Climate Agreement, the government, businesses, and societal organizations have agreed to disconnect all houses in The Netherlands from the natural gas grid by 2050 [23].

### 2.3.3. Finland

As in Germany, the FiTs for renewable energy were replaced with a premium system from the beginning of 2019. The new system is technologically neutral, and those renewable energy plants that offer electricity at the lowest premiums will be accepted into the system.

No biogas projects were proposed to the authorities in the auction in 2018 (1.4 TWh in total), and all projects that were accepted into the premium program use wind power [36]. New auctions have not been announced.

Biogas production is currently supported through two separate investment subsidy programs—one for industrial and another for agricultural plants. The investment support for large-scale industrial plants is paid by the Ministry of Employment and Economy. A maximum of 30% of the acceptable investment costs is covered.

The investment support for agricultural plants is paid from the EU Rural Development Programme 2014–2020 by the Ministry of Agriculture and Forestry. Farm-scale plants, which mainly use the energy themselves and do not sell any energy other than electricity outside the farm, are eligible for an investment subsidy of up to 40% of the acceptable investment costs. It is also possible to get the investment subsidy when most of the energy is sold outside the farm, or the farm sells traffic fuel. However, a separate company must be founded for this purpose. The agricultural company can then get an investment subsidy of up to 30% of the investment costs.

In addition, to support biogas production, the use of biomethane as traffic fuel has been exempted from fuel tax. However, the taxation of biomethane as traffic fuel is currently under discussion. The taxation would allow using the biofuel-blending obligation for biomethane in traffic gas, i.e., the natural gas that is sold as traffic gas would contain a certain amount of biomethane. On the other hand, the farms selling only biomethane could not benefit from this, but instead, the demand could decline as the price increases. Even without tax, biomethane is ca. 20% more expensive than natural gas for the consumer.

Finland's Integrated Energy and Climate Plan [4] has the following main targets by 2030:

- to reduce GHG emissions in the non-emissions trading sector by 39% (compared to 2005);
- renewable energy share of final energy consumption at least 51%;
- renewable energy share of final energy consumption 30% in road transport.

In addition, the plan has the following directly or indirectly biogas-related targets:

- to phase out the use of coal for energy production with minor exceptions;
- to decrease the domestic use of imported oil by 50%;
- to make electricity and heat production nearly emissions-free while also considering the perspectives of security of supply;
- have a minimum of 250,000 electric and 50,000 gas-driven passenger cars on the roads.

Some of these targets were already set by the former government in 2015, such as to have 50,000 gas-driven passenger cars by 2030 [37]. Another biogas-related target was to process 50% of all manure (e.g., in biogas production) by 2025 [38].

Regarding biofuels and biogas, these targets are still far away from the present situation. In 2019, the share of biofuels in road transport was 11% [22], and there were about 9400 gas-driven passenger cars in use, of which 3800 were newly registered (Traficom 2020) [39]. Approximately 6% of manure is currently processed [40]. Both the Ministry of Economic Affairs and Employment and the Ministry of Agriculture and Forestry are working with the implementation of the biogas-specific targets (Ministry of Economic Affairs and Employment 2020) [41]. The use of biomethane as traffic fuel is hoped to be encouraged with blending obligation. On the other hand, the use of manure as raw material for biogas production is planned to have an additional support tool.

## 3. Materials and Methods

The research data were gathered through three separate surveys in three EU member states, namely, Finland, Germany, and The Netherlands. The questionnaire was first prepared for the Finnish expert community. The survey design and content were planned through a pre-interview round in 2017 [14].

The questionnaire was divided into two parts and several sections. Part I included questions related to the business environment of renewable energy production in general and contained two sections, namely, (A) development of energy policy until 2030, and (B) development of business environment until 2030. Part II included biogas-specific questions and contained four sections, namely, (A) increasing and/or decreasing factors for biogas production and use, (B) income development for centralised biogas plants, (C) income development for farm-scale biogas plants, and (D) significance and roles of biogas technology in the future. The respondents were asked to evaluate statements from the present day to the year 2030 in mind. In Part I, only probable future was considered, whereas in Part II, both probable and desirable futures were included.

The range of responses used a seven-step Likert scale. Answers in Part I varied from totally disagree to totally agree ($-3$ = totally disagree, $-2$ = disagree, $-1$ = slightly disagree, $0$ = do not agree or disagree, $1$ = slightly agree, $2$ = agree, and $3$ = totally agree), and answers in Part II varied from decreases significantly to increases significantly ($-3$ = decreases significantly, $-2$ = decreases, $-1$ = decreases slightly, $0$ = stays on the current level, $1$ = increases slightly, $2$ = increases, and $3$ = increases significantly). It was also possible to answer "I cannot say" or not to answer at all.

The link to the online questionnaire (Webropol) was sent by email to the chosen respondents (in Finland in November 2018, in Germany in January 2020, and in The Netherlands in June 2020), and the survey was open for about one month. The questionnaire was sent to biogas producers in farm-scale and industrial-scale plants, technology suppliers, consultants, researchers, and policymakers/administration in the biogas field. The idea was to have an extensive and well-balanced expertise coverage within the biogas value chain.

Reminders were sent to the respondents, and a total of 84 responses (Finland 21 responses, Germany 41 responses, and Netherlands 22 responses) were received. Not all respondents answered all questions. Especially in Finland, only 11 respondents out of 21 answered also to biogas-specific questions in Part II. The reason for this was that the original Finnish survey was sent to a wider group of renewable energy experts which were not all specialists in the biogas field. In The Netherlands, 20 respondents out of 22 answered also to Part II, and in Germany, all respondents answered both parts of the survey. Moreover, not all respondents answering Part II also answered the questions related to the desirable future. In Finland, 9 respondents, in The Netherlands, 18 respondents, and in Germany, 40 respondents answered all questions related to both the probable and desirable future in Part II. Even in addition to that, few individual questions were left unanswered by some respondents.

The respondents' expertise was evaluated through background questions. In all countries, the degree of education was asked. Most of the respondents had at least a higher professional education (HBO) or university degree (Germany 93%, The Netherlands 100%, and Finland 86%). In Germany and The Netherlands, the field of education was also asked. Most of the respondents had been studying technology or natural sciences (Germany 80% and The Netherlands 77%) and most of the remainder were studying economics.

In Germany and The Netherlands, experience in the biogas field in years and professional background were also asked. In Germany, 51% of the respondents had less than 10 years' experience in the biogas field and 49% more than 10 years. In The Netherlands, 32% of the respondents had less than 10 years' experience and 68% more than 10 years. For professional background, the respondents could choose farm-scale biogas producer, industrial-scale biogas producer, biogas technology supplier, consultant, researcher, policymaker/administration, or other. In Germany, many of the respondents were researchers (56%). In The Netherlands, a large group of respondents had identified themselves as 'other' (32%). These are most likely people representing some biogas-related association because the survey was sent to several associations. However, the respondents' expertise covered different parts of the biogas value chain well. Furthermore, the aim of an expert

survey was not to have a statistically representative sample, but rather to reach different types of experts through theoretical sampling.

## 4. Results and Discussion

### 4.1. Part I: Business Environment of the Renewable Energy Production

A stable and predictable energy policy would encourage companies to do new investments accordingly and thus achieve the policy goals. However, most respondents in all three countries did not see the energy policy to be stable and predictable in the probable future (Figure 2, IA1). German respondents were most pessimistic followed by Finns and Dutch. This reflects the common problem with changing governments and fluctuating political decisions. Especially in Germany, fundamental changes were made to the EEG in 2017, when fixed FiTs were replaced by tenders. Yet, the respondents agreed on stricter climate policies and targets (IA2).

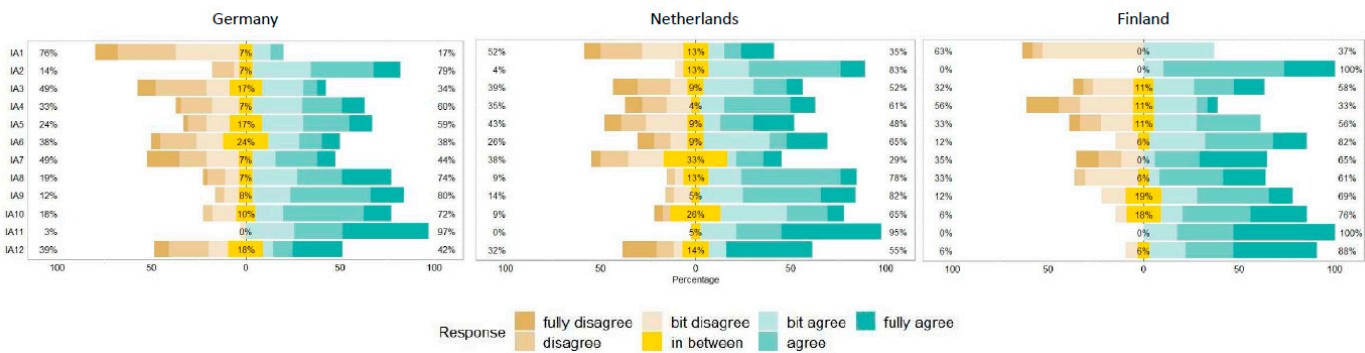

**Figure 2.** Development of energy policy until 2030 (probable future). IA1. The energy policy will be stable and predictable; IA2. climate policy and targets will become stricter; IA3. the best support form for renewable decentralised production is investment aid; IA4. the best support form for renewable decentralised production is long term production support; IA5. the best support form for renewable decentralised production is a combination of investment aid and production support; IA6. the best support form for renewable decentralised production is tax allowances; IA7. energy subsidies should be neutral in terms of scale and technology; IA8. the basis for energy subsidies should be energy efficiency; IA9. the basis for energy subsidies should be flexible production capacity; IA10. the basis for energy subsidies should be the capability of storing energy; and IA11. the basis for energy subsidies should be a reduction of GHGs; and IA12. any form of energy or fuel should not get permanent subsidies.

The opinion about the best support form for renewable decentralised energy production varied between countries (IA3–IA6). In Germany, 60% of the respondents chose long-term production support followed by a combination of investment aid and production support (59%). Likewise, in Germany, the biogas business branch is used to benefit from the long-term production support. Until the latest version of EEG 2017, the EEG offers 20 years of stability for the individual plant operator. This stability was the reason for the growth of the biogas branch in Germany.

In The Netherlands, most experts agreed with current support for renewable energy, which is mainly consisting of production support and also includes several investment support measures. However, tax allowances were the most popular among the experts (65%). From the question, it is not a priori clear which type of tax allowances are meant. In The Netherlands, the use of electricity that businesses and private households produce themselves from renewable sources (e.g., biogas, landfill gas, sewage gas, and electricity from CHP installations) is exempted from ODE tax. The exemption does not apply to energy consumption; renewable energy and grey energy have the same taxes for consumers.

In Finland, tax allowances were strongly favoured (82%), but investment aid was also seen positively (58%). Currently in Finland, the traffic use of biomethane is exempted from fuel tax. Otherwise, investment support is the leading support measure in Finland, both for the industrial and farm-scale biogas plants.

In all three countries, the reduction of GHGs was considered as the most important basis for energy subsidies (IA8–IA11). Likewise, the respective governments have announced further increases in goals for the reduction of $CO_2$ and ambitious plans to curb climate change. In Germany and The Netherlands, the flexible production capacity was the second popular option, and in Finland, the capability of storing energy. Especially in Germany, the strong growth in fluctuating wind and solar energy is causing a high demand for balancing energy supply, which will continue to rise sharply in view of the political expansion targets. Therefore, the flexibilization of biogas plants has been promoted since the EEG 2014 and is even a prerequisite for participating in the tenders of the EEG 2017.

Whether the energy subsidies should be neutral in terms of scale and technology (IA7), or any form of energy or fuel should not get permanent subsidies (IA12), divided the opinions both between and within countries. In Germany, neutrality in terms of plant size and technology was never intended under the EEG. The different FiTs for the different forms of energy, such as wind, solar, or biomass, were based on the financial needs of these technologies and their different plant sizes. In addition, bonuses were used to create incentives, e.g., for the use of certain substrates. Both in Germany and The Netherlands, the majority of the respondents were against scale- and technology-neutral subsidies. In Finland, scale- and technology-neutral subsidies were mainly supported (65%), and only Finns agreed strongly against permanent subsidies (88%).

The role of consumers as small producers of both electricity (IB1) and heat (IB2) was seen to become more common in all three countries in the probable future until 2030 (Figure 3).

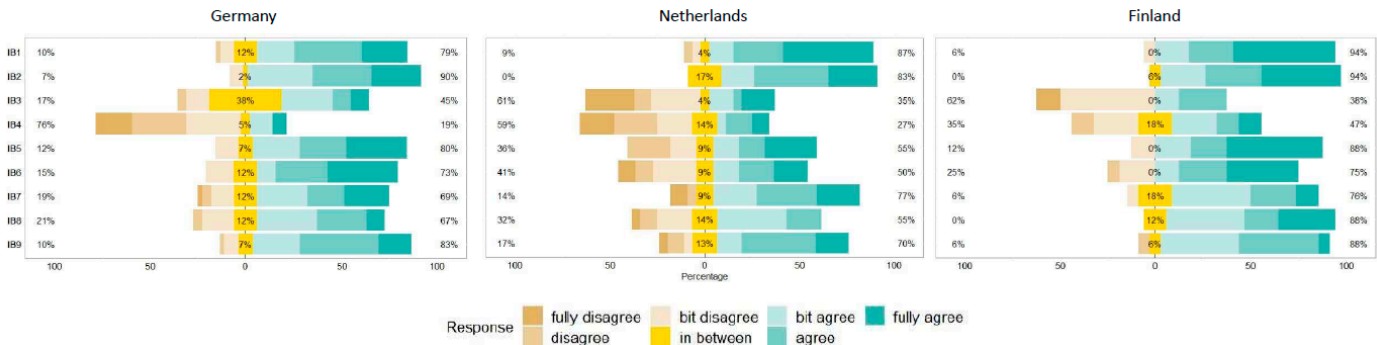

**Figure 3.** Development of business environment until 2030 (probable future). IB1. New technologies increase smart grids and enhance the role of consumers as small producers; IB2. various decentralised heat production technologies as part of wider heat grid become more common; IB3. energy production with wood chips is economically viable also without support; IB4. energy production with biogas is economically viable also without support; IB5. wind energy is economically viable also without support; IB6. solar energy is economically viable also without support; IB7. local energy companies and farms are co-investing for local renewable energy production; IB8. as the consumer awareness of the energy choices increases, their willingness to pay increases too; and IB9. the amount and quality of energy used in production becomes more relevant for the image of the consumer product (branding).

Renewable energy production without any support was seen most probably economically viable with wind energy (IB5: 55–88%) and second with solar energy (IB6: 50–75%). Wood chips (IB3) were mostly not considered economically viable without support, and biogas (IB4) even less. For wind and solar power, cost reduction has been achieved by technology development, but this has not been the case for biogas because a large part of the costs for biogas plants are the substrates (or their logistical costs), which are not subject to any technological development. Moreover, in Germany, increased administrative requirements and safety regulations included in the EEG 2017 compensated for the cost reduction achieved by technology development. Finns were most positive about the economic viability of biogas without support (47%) followed by Dutch (27%) and Germans (19%). One reason for this might be that, in Finland and The Netherlands, less energy crops

are used for biogas production and energy production itself has a smaller role besides waste treatment and nutrient recycling.

The respondents in all three countries believed in co-operation between farms and local energy companies in local renewable energy production (IB7: 69–77%). There also seems to be a great trust in consumers regarding the willingness to pay for environmentally friendly energy and their consumption behaviour. All these possibilities could also benefit the biogas business branch.

### 4.2. Part II: Biogas Specific Questions

The investment cost for a biogas plant was believed to increase in the probable future in Germany and Finland (56% and 50%) but not so much in The Netherlands (Figure 4: IIA1). The investment costs of biogas plants have increased over time due to the increased safety requirements. However, in the desirable future, most respondents in all countries hoped that the prices would decrease. The state of the biogas technology was believed to increase in all countries in the probable future (70–100%) and even more in the desirable future (IIA2).

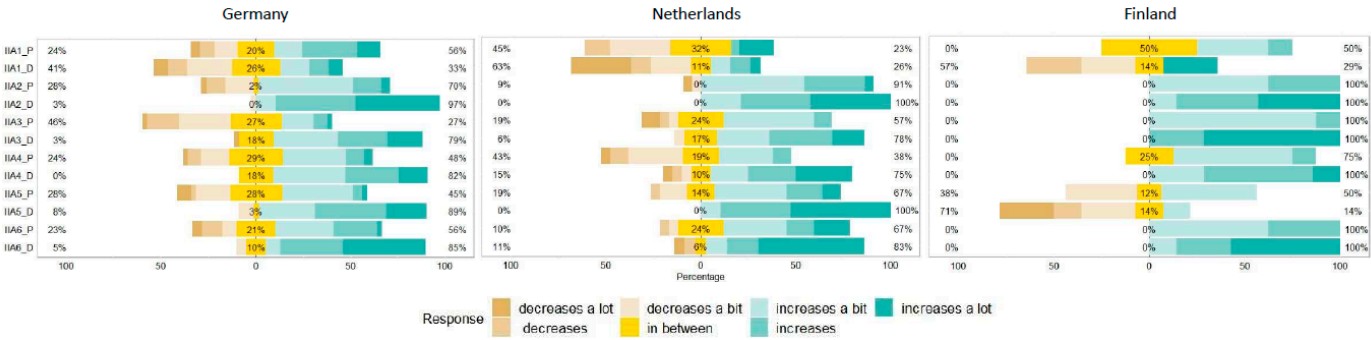

**Figure 4.** Increasing and/or decreasing factors for biogas production and use (P = probable, D = desirable future). IIA1. Investment cost of biogas plants; IIA2. state of the biogas technology; IIA3. availability of suppliers and technologies of farm-scale biogas plants; IIA4. availability of suitable raw materials for biogas production; IIA5. the price (that a biogas plant can receive) for recycled fertilisers (digestate-based); and IIA6. extension of biogas filling station network.

The Germans were pessimistic about the availability of suppliers and technologies of farm-scale biogas plants in the probable future (IIA3). Of the respondents, 46% believed that the availability would decrease. This might be because the EEG 2017 led to a sharp reduction in plant construction. As a result, some plant manufacturers had to file for insolvency. Suppliers of small liquid manure plants and suppliers with foreign business came through this crisis better. In The Netherlands and Finland, the availability of suppliers and technologies were believed to increase in the probable future, and in all countries, in the desirable future.

In The Netherlands, the availability of suitable raw materials for biogas production shared opinions in the probable future (IIA4). The somewhat unclear answer to this question may be because current production focuses more on waste and residual materials. This increases the number of substrates used, but not their availability. The potential for manure digestion is still very high. In Germany, the respondents favoured an increase in the raw material availability in the probable future (48%), although there are restrictions on the use of maize silage in the EEG 2017 (max. 44% from 2021). Also, in their research, Pehlken et al. concluded that bioenergy supply chains involving alternative biomass and grass from grasslands provide optimisation potentials compared to the current corn-based practice [42]. In Finland, the availability of raw materials was believed to increase strongly.

The question IIA5 was formulated in the first Finnish query as 'The price for recycled fertilisers'. Thus, in Finland, it could have been understood differently as the production cost for recycled fertilisers. In that way, it would be considered positive if the production costs would decrease, and the biogas plant could sell the fertiliser product at a lower

price and thus get a larger market share for their products. Otherwise, in Germany and The Netherlands, the price that a biogas plant can receive for recycled fertilisers were believed to increase in the probable future (45–67%). In practice, the achievable prices for digestate-based recycled fertilisers vary greatly from region to region. In areas with a lot of livestock farming and a high density of biogas plants, there is a large surplus of digestate. The processing of the digestate to a commercial fertiliser is cost-intensive and is mainly practiced when the nutrient surplus of the region is so high that a long transport becomes necessary.

The biogas filling station network (IIA6) was believed to extend in all case study countries in the probable future, although in Germany (56%) less than The Netherlands (67%) and Finland (100%). The demand for gas-driven passenger cars is very low in Germany. Only 1% of passenger cars are powered by natural gas or biogas. The utilisation of biogas as a fuel source is almost exclusively realised by feeding it into the natural gas grid and withdrawing it from the balance sheet at natural gas filling stations. The slightly positive tendency regarding the development of natural gas filling stations may be due to the decreasing attractiveness of using biogas for electricity generation. In addition, the fuel sector, through REDII, will have to use more renewable fuels in the future. In Finland, the current and former governments have strongly supported biogas traffic use.

Respondents in all case study countries believed that the price of the competing fuel for biogas (natural gas) would increase in the probable future (Figure 5, IIA7: 57–60%), as well as the availability of suppliers and technologies of small-scale purification units for traffic gas production (IIA8: 62–86%). However, the number of such suppliers is so small that a decrease would not be possible. There is currently no supplier of small turnkey upgrading plants on the German market. However, the components are available, and there are suppliers of natural gas filling stations.

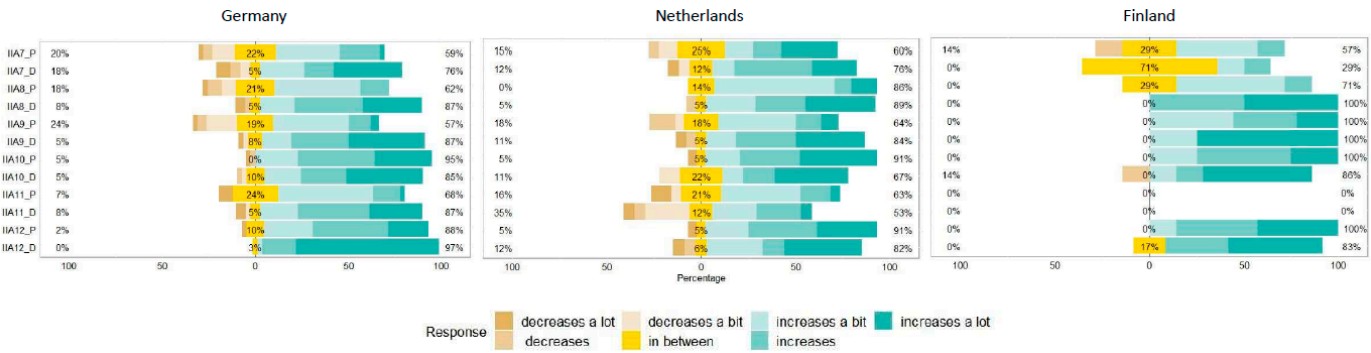

**Figure 5.** Increasing and/or decreasing factors for biogas production and use (P = probable, D = desirable future). IIA7. Price of the competing fuel for biogas (natural gas); IIA8. availability of suppliers and technologies of small-scale purification units for traffic gas production; IIA9. number of gas cars; IIA10. number of electrical cars; IIA11. number of hydrogen cars (lacking from Finnish survey); and IIA12. favouring low emission cars in the cities (Germany, Finland)/emission limit for city traffic (The Netherlands).

Interestingly, the respondents in all countries wanted to believe that the number of gas cars would increase in the probable future (IIA9: 57–100%), although the trend is seen very differently by the car manufacturers which clearly are gradually backing off from the further development of gas-driven passenger cars. Maybe the reasons for the optimistic rating can be found in heavy-duty vehicles and in RED II. However, in Finland, the number of gas-driven passenger cars is still increasing.

The number of electrical cars was believed to increase even more than gas-driven cars in the probable future (IIA10: 91–100%), which is no surprise. The public focus is strongly on electric cars. The question about the number of hydrogen cars was lacking from the Finnish survey. In Germany and The Netherlands, the respondents believed that the number of hydrogen cars would increase as much in the probable future as natural

gas or biogas cars (IIA11: 63–68%). However, in The Netherlands, the experts' opinions about hydrogen cars were divided on what comes to the desirable future. Currently, the availability of filling stations and cars is so low that it is not a real alternative. Nevertheless, both German and Dutch governments have strong hydrogen strategy.

Favouring low emission cars in the cities or emission limits for city traffic will most probably increase in all countries (IIA12: 88–100%). However, the current debate is more about air quality than renewable fuels and GHG emissions.

Centralised plants were referred here as large-scale industrial plants which can use various waste or side streams or agricultural biomasses as raw materials either separately or in co-digestion. Income from production and selling of biomethane for traffic use was seen to be increasing most regarding the centralised biogas plants in all countries, both in the probable and desirable future (Figure 6, IIB5). In Germany and The Netherlands, the two other most increasing income options were selling gas for consumers and industry (IIB4) and heat production (IIB2).

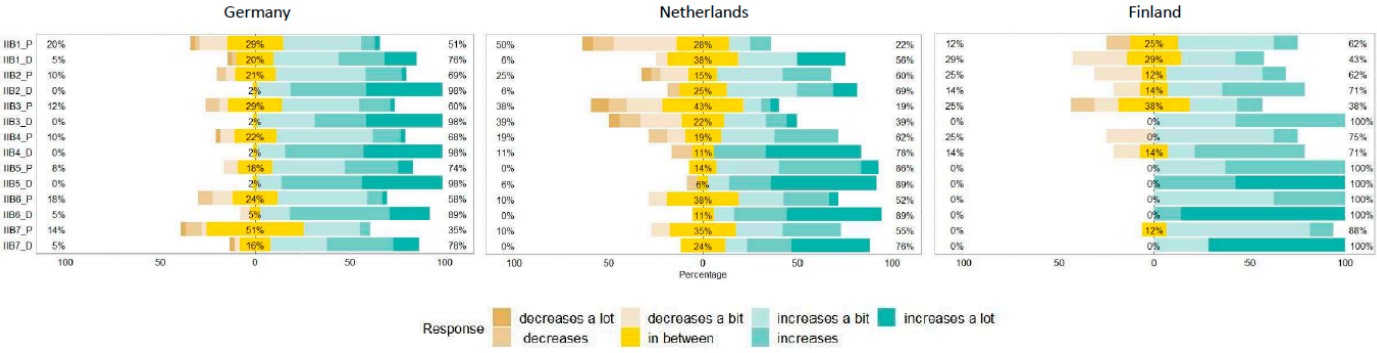

**Figure 6.** Income development for centralised biogas plants (P = probable, D = desirable future). IIB1. Income from gate fees; IIB2. income from heat production; IIB3. income from combined heat and power production (CHP); IIB4. income from selling gas for consumers and industry (using the gas grid); IIB5. income from production and selling of biomethane for traffic use; IIB6. income from recycled fertilisers; and IIB7. income from biochemicals.

In Finland, recycled fertiliser products and biochemicals were seen to increase most after the biogas traffic use. Biochemicals could have got more votes also in Germany and The Netherlands if the timeframe would have been longer. Maybe many respondents thought that 2030 is too close and biochemicals are still far away from being a common technology.

Despite the preferences, most income options were believed to increase. However, in The Netherlands, most respondents believed that the income from gate fees (IIB1) and CHP production (IIB3) would decrease in the probable future. Already in the current business environment, gate fees are only important for waste plants. The economic feasibility in CHP production is challenging both because of the low electricity price and high maintenance costs of the CHP unit, ca. 0.013 €/kWh$_{el}$ [43]. A further shift towards biomethane and transport fuels, to the disadvantage of the production of electricity and heat from biogas, might be expected.

Farm-scale plants were referred here as single farm plants using mainly agricultural biomasses. Branding agricultural products with carbon-neutral labels (Figure 7, IIC6) was most highly rated in Germany, maybe because it is not common yet. Also, in The Netherlands and Finland, most respondents believed that this will increase strongly. As one example from Finland, Valio Ltd. is aiming for carbon-neutral milk production by 2035, and part of the solution is using cow slurry as biogas raw material and furthermore using biomethane as fuel for the tank trucks collecting milk [44].

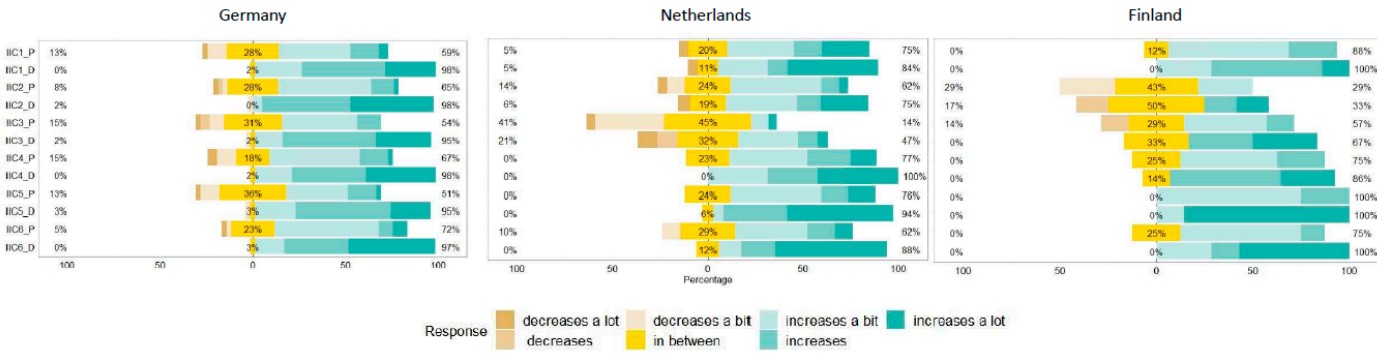

**Figure 7.** Income development for farm scale biogas plants (P = probable, D = desirable future). IIC1. Improving farm energy self-sufficiency; IIC2. income from selling energy out of the farm—heat production; IIC3. income from selling energy out of the farm—CHP production; IIC4. income from selling energy out of the farm—traffic gas purification; IIC5. improving nutrient self-sufficiency and crop yields; and IIC6. branding agricultural products with a carbon-neutral label (and thus getting higher price).

Both in The Netherlands and Finland, the largest growth potential for income development was believed to be in improving nutrient self-sufficiency and crop yields (IIC5). Biogas plant digestate is a better fertiliser than manure because during AD part of the organic nitrogen is degraded into ammonium nitrogen, which is directly available for the plants. Especially in organic farming, the nutrient use can be improved with biogas plant digestate instead of direct use of animal manure or green manure, i.e., grass or legumes cultivated as part of crop rotation [45].

In Germany, the second-largest income growth potential was seen in selling energy out of the farm in the form of traffic gas (IIC4), which was also highly rated in The Netherlands and Finland. However, Dutch and Finns preferred improving farm energy self-sufficiency as the second option (IIC1). In Finland, there is a clear reason for this. Farms are remotely located and district heating is not possible. Thus, at least some own heat production is needed. The most common heat source is wood chips, but biogas is also a viable option. The only clear decreasing income source was, according to Dutch respondents, selling energy out of the farm in the form of heat and electricity, i.e., CHP production (IIC3). Also, the Germans and Finns saw this among the least promising options. In Finland, the least favourable option was selling energy out of the farm in the form of heat (IIC2), which describes the current situation well. Because of the long distances between farms and other settlements, it is typically not possible to sell heat out of the farm in Finland.

The respondents in all countries were very positive about the many important roles of biogas technology in the future. Treatment of manure and cutting down emissions from agriculture (Figure 8, IID6) got the highest ranking both in the probable and in the desirable future. Also, economically, manure digestion is an extra option with higher support in Germany.

The second most important role was seen in nutrient recycling (IID8). Unlike in composting, where nitrogen is lost through denitrification in the form of nitrogen gas to the atmosphere, nitrogen compounds remain in the digestate, and their fertilisation effect is even improved. Likewise, AD suits well for the treatment of biowaste (IID7). The moisture content does not harm the process as in combustion and the process enables nutrient recycling.

The third-most important role seen by the experts was reducing $CO_2$ emissions from traffic and improving urban air quality (IID2). These challenges indeed provide great potential for improvement. The transport sector represented 25% of the GHG emissions in 2018 within EU-27 member states [46]. Many European countries are currently promoting electric vehicles (EVs) as a leading GHG mitigation solution for the transportation sector. However, EVs have a high level of production-related emissions. The emissions from their use, on the other hand, depend on the GHG intensity of the electric grid in question. In

the worst case, EVs can lead to greater life-cycle GHG emissions than comparable diesel vehicles. The probability that an EV will lead to lower life-cycle GHG emissions than a diesel vehicle is only 75% for Germany and The Netherlands, whereas for Finland it is 99% because of the high share of renewables in electricity production. However, there are several countries in the European Economic Area (EEA), such as Poland, Latvia, and Estonia, where the emissions from EVs most probably exceed those from diesel vehicles [47]. The use of biomethane as a vehicle fuel has one the lowest well-to-wheels GHG emissions, comparable to the use of renewable electricity for EVs, according to the latest JEC Well-To-Wheels report [48]. Unfortunately, the current $CO_2$ standards for car manufacturers do not recognise biomethane, but the $CO_2$ emissions for gas-driven cars are calculated based on natural gas. The European Commission will propose a revision of the $CO_2$ standards for cars and vans by June 2021 and will also review the $CO_2$ standards for heavy-duty vehicles by 2022 [49]. To be able to meet the aims of the EU's 'Sustainable and Smart Mobility Strategy', i.e., a 90% reduction in transport-related GHG emissions by 2050, the policies should take into account the emissions from the whole life-cycle of a vehicle. In addition, a solution based on several technologies and energy sources would be more resilient and enables selecting the best solution according to the local conditions and needs.

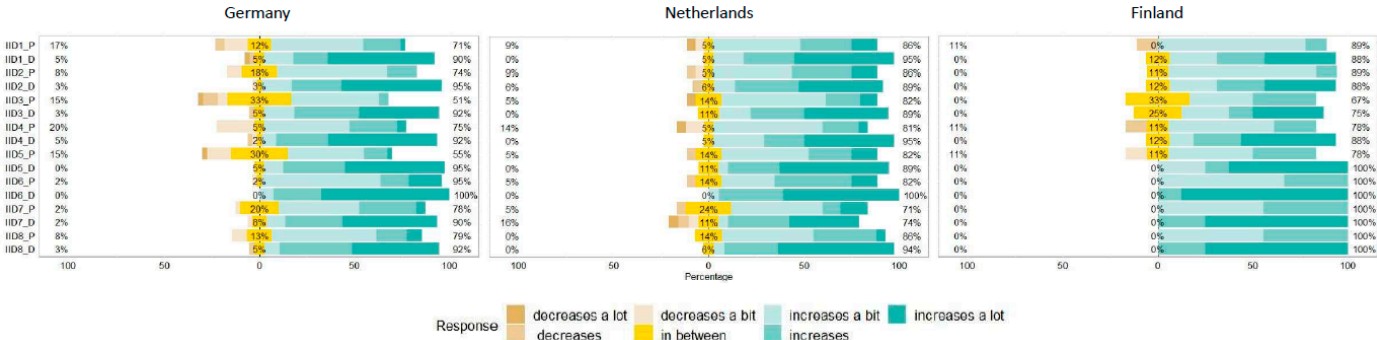

**Figure 8.** Significance and roles of biogas technology in the future (P = probable, D = desirable future). IID1. Replacing fossil raw materials in the production of energy and fuels; IID2. reducing $CO_2$ emissions from traffic and improving urban air quality; IID3. replacing fossil raw materials in the production of chemicals and materials; IID4. using biogas for balancing production and storing energy along with other renewable energy sources (e.g., wind, solar); IID5. biogas as part of the decentralised energy production for improving the security of supply; IID6. treatment of manure and cutting down emissions from agriculture; IID7. treatment of other types of biowaste; and IID8. recycling nutrients.

## 5. Conclusions

The three case study countries had different approaches relating to biogas due to political aims and local conditions. In Germany, biogas has been used for electricity production based on generous FiTs. At present, with the tendering model and lower subsidy levels, research and practice are discussing ways to improve the economic efficiency of biogas plants, such as the use of cost-effective substrates, improved heat utilisation, improved utilisation of digestate as fertiliser, ecosystem services, flexible power production for grid stabilisation or demand-driven production (which is a requirement to enter the tender). However, alternatives without state subsidies are also being sought. One possibility could be biomethane production for gas grid injection or use as traffic fuel. Most of the approaches currently show no economic viability or require the removal of legal hurdles for their implementation. How serious the situation is can be seen from the fact that 2020 is the first year in which the number of biogas plants in Germany decreases.

In The Netherlands, natural gas has been used widely, both in households and by the industry, and the country has a wide gas grid. Due to safety issues, the use of natural gas has been decided to phase down by 2050 and be replaced by biomethane and hydrogen gas. Biomethane could also find more use in traffic, especially heavy-duty vehicles and inland shipping. The current biogas production could be more than doubled only by using

manure as a substrate due to the high density of livestock. Biogas production is growing steadily, and biomethane injection to natural gas grid is growing fast.

In Finland, the biogas business branch is challenged with long distances and small markets. Biogas is hoped to give a new income source for agriculture by combining food and energy production, and at the same time, cut down emissions. Both the former and present Finnish governments have supported both biogas production and the use of biomethane as traffic fuel. Biogas production in Finland is still small, measured both in absolute production amount and per capita, but especially, the use of biomethane as traffic fuel is growing fast.

Despite the differences, the biogas business branch in the case study countries also had common drivers and barriers based on expert survey. In the following Table 2, the drivers and barriers are categorised based on a PESTE/PESTLE framework [50,51].

**Table 2.** The drivers and barriers of biogas production based on expert survey.

| PESTE | Drivers | Barriers |
|---|---|---|
| **P**olitical | - Renewable energy and GHG emission reduction goals<br>- Governmental support mechanisms | - Uncertainty of political decisions<br>- Business models based on subsidies<br>- Biomethane not recognised in $CO_2$ standards for car manufacturers |
| **E**conomical | - New income sources from traffic fuel and circular economy solutions (recycled nutrients and biochemicals)<br>- Branding agricultural products | - High logistical costs of raw materials<br>- Low price of end products due to the low price of fossil energy and mineral nutrients<br>- Underdeveloped markets for recycled nutrients |
| **S**ocial | - Significance of 'Green consumers' increases<br>- Co-operation between farms and local energy companies<br>- Decentralised energy production creates jobs for rural communities | - Partly misleading negative images about biogas production ('not in my backyard' phrase)<br>- Too positive images about electrical cars |
| **T**echnological | - Increase in the number of technological solutions regarding biogas plants and biomethane up-grading | - Lack of interest from car manufacturers to develop new gas-driven vehicles |
| **E**nvironmental | - Emissions' reduction from agriculture<br>- Nutrients recycling<br>- Replacing fossil fuels | - Emissions from biogas plants and digestate storages |

Finally, how did the experts see the future role and opportunities for biogas in the three case study countries? The most important role was considered to be the treatment of manure and cutting down emissions from agriculture. The next important role was enabling nutrient recycling, and the third, reducing $CO_2$ emissions from traffic and improving urban air quality, although they were all highly rated and the differences between these options were small.

To be able to accomplish these tasks, the biogas business must be profitable. But how chould the businesses find new income sources or how could the policymakers create a stable business environment? The promising income sources were different for centralised and farm-scale plants. For centralised plants in all countries, the income from production and selling of biomethane for traffic use was seen to be increasing most. This could indeed be a win-win option for both the environment and the biogas branch. However, the future of this path lies in the policymakers' hands. How biomethane is rated in the future, in terms of $CO_2$ standards and in comparison to electrical vehicles, will be a central question regarding the biomethane use as traffic fuel.

For farm-scale plants, branding agricultural products with a carbon-neutral label was most highly rated in Germany. Also, in The Netherlands and Finland, there seemed to be a great trust in consumers' willingness to pay for food produced with environmentally friendly energy or with fewer emissions. However, both in The Netherlands and Finland, the largest growth potential for income development was believed to be in improving nutrient self-sufficiency and crop yields.

The expert survey showed the versatile roles and opportunities of biogas technology for societies. While the EU aims to be carbon neutral by 2050, biogas technology could help to achieve this goal by contributing to many sectors of the economy.

**Author Contributions:** Conceptualization, E.W., P.R., M.V.G., D.O. and S.Z.; Methodology, P.R. and M.V.G.; Formal Analysis, D.O. and A.G.; Investigation, E.W., P.R., M.V.G., D.O. and S.Z.; Writing–Original Draft Preparation, E.W., M.V.G., S.Z., P.R., D.O., L.Z., A.G.; Writing–Review & Editing, E.W.; Visualization, E.W. and D.O. All authors have read and agreed to the published version of the manuscript.

**Funding:** The Finnish study was part of the FutWend-project: Towards a future-oriented 'Energiewende', funded by the Academy of Finland (grant number 297747).

**Institutional Review Board Statement:** The survey study follows the regulations of the National Advisory Board on Research Ethics (2009) and the American Psychological Association, including the informed consent, confidentiality and anonymity of the participants.

**Informed Consent Statement:** Informed consent was obtained from all subjects involved in the study.

**Data Availability Statement:** Data available on request due to that the surveys were conducted in German, Dutch and Finnish.

**Acknowledgments:** We gratefully acknowledge the experts in all three case study countries who answered the Webropol survey and gave their time and expertise to our use.

**Conflicts of Interest:** The authors declare no conflict of interest. The founding sponsors had no role in the design of the study; in the collection, analysis, or interpretation of data; in the writing of the manuscript, and in the decision to publish the results.

## Abbreviations

| | |
|---|---|
| AD | anaerobic digestion |
| CBG | compressed biogas |
| $CH_4$ | methane |
| CHP | combined heat and power |
| $CO_2$ | carbon dioxide |
| EEG | Erneuerbare Energien Gesetz |
| FiT | feed-in tariff |
| GHG | greenhouse gas |
| GWh | gigawatt-hour |
| kWh | kilowatt-hour |
| NECP | national energy and climate plan |
| $Nm^3$ | normal cubic meter |
| RED II | Revised Renewable Energy Directive |
| SDE | Stimulering duurzame energieproductie en klimaattransitie |
| SNG | synthetic natural gas |

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
