# Peer review of "Expert Views on the Future Development of Biogas Business Branch in Germany, The Netherlands, and Finland until 2030"

_sustainability, doi:10.3390/su13031148_

Round 1

Reviewer 1 Report

The paper was revised according to the journal rules. 

The main topic treated consider the biogas exploitation in three european countries. Few revisions are required and reported below:

  • please add a nomenclature section with unit of measure
  • check and correct subscripts and superscripts
  • A graphical abstract should be added
  • More keywords should be added 
  • Abstract in the last part should be focused on results achieved 
  • Check the reference style 
  • Please add a reference in (Moreover...locally)
  • Check that in section 2 the proper references are cited
  • Figure 1 should be revised maybe combining them in online one figure 
  • Maintenance aspects should be deeply considered in the manuscript
  •  

Reviewer 2 Report

The topic of the article is current and important. It is an interesting case study of three countries in terms of the current state and prospects of biogas branche - against the background of national and EU regulations.

Due to the clear and generally technically correct language and the popular-science character, it may be interesting for a very wide audience, especially for decision-makers for whom it may constitute an important premise for the selection of future biogas development strategies and support systems.

Data for the evaluation of existing system solutions on the biogas market and conclusions for the future were collected on the basis of carefully designed surveys, sent to selected biogas experts (producers, investors, designers, researchers, politicians / decision makers, ...). Despite the fact that the scope of these surveys was relatively narrow (only 84 responses from 3 countries in total), it seems that the expert opinions obtained and the conclusions drawn based on them reflect the actual state of affairs and the desired trends.

In my opinion, the most important achievements of the research include defining barriers to the development of the biogas branch, including that biomethane is not recognized in CO2 standards for car manufacturers. By the way, the authors rightly noticed that electric cars are perceived too positively.

In addition, they indicated manure management as an important premise for the development of biogas plants, in addition to reducing emissions.

Specific comments:

Page 1:

The second sentence in Introduction need reference.

Page 9, after "main targets by 2030:"

it seems a more appropriate form is "to reduce ..." and lowercase spelling as below.

Page 13:

Instead of "One reason for this might be than ..." should be: "One reason for this might be that ...".

Page 15:

Instead of "natural or biogas" should be "natural gas or biogas".

Page 16

As above: "natural/biogas"

Table 2 - what does "PESTE" mean?
